# Dry Eye in Systemic Sclerosis Patients: Novel Methods to Monitor Disease Activity

**DOI:** 10.3390/diagnostics10060404

**Published:** 2020-06-13

**Authors:** Caterina Gagliano, Elisa Visalli, Mario Damiano Toro, Roberta Amato, Giovanni Panta, Davide Scollo, Giovanni Scandura, Salvatore Ficili, Giorgio Amato, Alessia Benenati, Roberta Foti, Giulia Malaguarnera, Giuseppe Gagliano, Raffaele Falsaperla, Teresio Avitabile, Rosario Foti

**Affiliations:** 1Ophthalmology Clinic, San Marco Hospital, Catania University, 95 123 Catania, Italy; robertaamato@nestweb.it (R.A.); giovannipanta@msn.com (G.P.); davidescollo@hotmail.com (D.S.); giovanni.scandura@hotmail.it (G.S.); salvo.ficili@icloud.com (S.F.); t.avitabile@unict.it (T.A.); 2Neurovisual Science Technology (NEST) srl, 95123 Catania, Italy; giuseppe.gagliano99@gmail.com; 3Rheumatology Unit, San Marco Hospital, Policlinico University of Catania, 95123 Catania, Italy; elivisa21@gmail.com (E.V.); giorgioamato@hotmail.it (G.A.); alessia.benenati@libero.it (A.B.); rosfoti5@gmail.com (R.F.); 4Department of General Ophthalmology, Medical University of Lublin, 20079 Lublin, Poland; 5Faculty of Medical Sciences, Collegium Medicum Cardinal Stefan Wyszyñski University, 01815 Warsaw, Poland; 6Faculty of Medicine, University of Catania, 95123 Catania, Italy; robertafoti@hotmail.com; 7Department of Biomedical and Biotechnological Science, University of Catania, 95123 Catania, Italy; giulia.malaguarnera@live.it; 8Neonatal Intensive Care, San Marco Hospital, 95123 Catania, Italy; raffaelefalsaperla@hotmail.com

**Keywords:** dry eye syndrome, systemic sclerosis, skin score, meibomian gland disease, lipid tear dysfunction, tear osmolarity, Schirmer test, break-up time test

## Abstract

Background: In systemic sclerosis (SSc) patients, dry eye syndrome (DES) is the most frequent ocular feature. The aim of this study was to investigate ocular DES-related SSc patients and to establish any correlation with the severity of the disease. Methods: Retrospectively, data from 60 patients with SSc underwent ophthalmic examination, where non-invasive film tear break-up time (NIF-TBUT), tear film lipid layer thickness (LLT), anesthetic-free Schirmer test I, tear osmolarity measurement (TearLab System), and modified Rodnan skin score (mRSS) data were collected. The visual analog scale (VAS) and Symptom Assessment in Dry Eye (SANDE) methods were utilized. The results were correlated with mRSS and the duration of SSc. Results: Severe DES occurred in 84% of cases, and was more severe in women. The eyelids were involved in 86.6%, secondary to meibomian gland disease (MGD). A direct correlation was found between the tear osmolarity (mean 328.51 ± 23.8 SD) and skin score (mRSS) (r = 0.79; *p* < 0.01). Significantly reduced NIF-TBUT, LLT, and Schirmer test I values were observed in the case of severe skin involvement. Conclusions: SSc patients show lipid tear dysfunction related to the severity and duration of the disease due to inflammation and the subsequent atrophy of the meibomian glands.

## 1. Introduction

Systemic sclerosis (SSc; scleroderma) is a complex multisystem autoimmune disease of unknown etiology, characterized by vasculopathy and tissue fibrosis of the skin and various internal organs [1,2]. Genetic and environmental factors determine the susceptibility, severity, and onset of this disease [1]. Its peak incidence is between 30 and 60 years of age with a female predilection (6:1 ratio), while a more severe expression of the disease, including internal organ-based complications and higher mortality, has been reported in men [3,4].

The clinical manifestations of the disease are the result of three distinct processes: innate and adaptive immune system abnormalities, leading to the production of autoantibodies and cell-mediated autoimmunity, vasculopathy of small vessels, and fibroblast dysfunction leading to the accumulation of excessive collagen and other matrix components in the skin, blood vessels, and internal organs [5,6,7].

Dry eye syndrome (DES) is a dysfunction of the tears and the ocular surface; it is very common and complex, affecting 5–50% of patients, giving rise to symptoms of discomfort, visual disturbances, and tear instability due to an increase in tear osmolarity and the inflammation of the ocular surface [8,9]. Due to increases in video terminal syndrome and office eye disease syndrome, the prevalence of DES rises annually. Extensive research and the publication of diagnostic and treatment guidelines related to the disorders that cause DES [10,11,12] have led to the availability of new diagnostic techniques and therapeutic options in patients affected by DES.

It has been shown that patients with systemic autoimmune disease have a morbidity of 51.7% for dry eye, which is noticeably higher than that of the control group [13]. This may be caused by an imbalanced regulatory mechanism of the protective immunity on the ocular surface [14]. When the ocular immune system is excessively stimulated and/or the immunoregulatory mechanisms are disrupted, the balance between the innate and adaptive phases is dysregulated and chronic ocular surface inflammation can result, leading to chronic DES [15].

Additionally, changes in collagen content and distribution within the cornea, the transparent outer layer of the eye and the main refractive unit, may cause plausible pathologic changes to ocular function in SSc patients. Indeed, in SSc patients, among the most common reported ocular pathologies is eyelid stiffness, which is seen in more than 50% of cases and results from the deposition of type I collagen in the dermis [4]. Keratoconjunctivitis sicca is the second most common problem, seen in 50% of affected patients, followed by cases of conjunctivitis, episcleritis, anterior uveitis, and hypertensive retinopathy [16].

To date, while a multisystemic involvement is well and widely documented in SSc patients, only single case reports or small case studies regarding ocular involvement in SSc patients have been reported in the literature, and further evidence is demanded [8].

The aim of this study was to explore the prevalence and characteristics in the ocular symptoms and signs of dry eye in patients with SSc and to establish any correlation with the severity of the disease.

## 2. Materials and Methods

In this retrospective study, we included all consecutive patients affected by SSc and diagnosed, according to the corresponding international criteria [17,18], at the Rheumatology and Ophthalmology Units, San Marco Hospital of Catania (Italy) from 3 February 2016, to 5 December 2019.

The study protocol, approved by the Local Ethics Committee, conformed to the tenets of the Declaration of Helsinki. A written informed consent form for the processing of personal data was obtained from all patients.

All included patients underwent a serological test to rule out Sjogren’s syndrome, and a clinical history of secondary Sjogren’s syndrome was excluded during the entire follow-up period [19].

Figure 1 shows the workflow of the study.

All patients underwent a full ophthalmic examination using a slit lamp and an indirect binocular ophthalmoscope.

An Early Treatment Diabetic Retinopathy Study (ETDRS) chart was used to test the best-corrected visual acuity (BCVA).

Intraocular pressure (IOP) was assessed using an I-care^®^ tonometer (Tiolat Oy, Helsinki, Finland, EU).

The Schirmer test (type I) without anesthetic [15], non-invasive tear break-up time (NIF-TBUT), tear osmolarity (Tear Lab), and tear film lipid layer thickness (LLT) tests were carried out following the indications of Tashbayev et al. [20]. The anesthetic-free Schirmer test (type I) was performed using standardized filter paper strips (Alcon Laboratory, Fort Worth, TX, USA) [19].

A TearLab^®^ system (Tear-Lab Corporation, San Diego, CA, USA) was used to measure osmolarity. A sample of approximately 50 nL of tears was collected from the lower tear meniscus to obtain a reading, according to the manufacturer’s recommendations [19].

Meibography and a graduate scale of glandular tissue loss area were used to study meibomian gland disease (MGD) abnormalities. The inflammation of the meibomian glands was assessed by observing the alteration of the eyelid margin, the thickening of the Marx line, and the morphological alteration of the glandular bodies in the meibographs.

The tear film break-up time (TFBUT) was assessed non-invasively, without fluorescein, by using a keratograph^®^ (CSO, Scandicci, Italy, EU) [21]. For the non-invasive film tear break-up time (NIF-TBUT) test, a value in the range of 20–30 s was considered normal, while values below 10 s as pathological.

LLT, blinking pattern and meibomian gland images were assessed using the Lip iView II (TearScience, Morrisville, NC, USA) ocular surface interferometer.

Evaluation of the symptoms was made using visual analog scale (VAS) and the Symptom Assessment in Dry Eye (SANDE) questionnaires. To quantify the patients’ symptoms of ocular dryness and/or irritation (i.e., foreign body sensation, burning/tingling, itching, pain, sticking sensation, blurred vision, and photophobia), a unidimensional VAS, 10 cm in length (equivalent to 10 degrees) was used, with its numbers (degrees) being visible only on the side of the examiner. The intensity of the discomfort was indicated with a vertical line on the line between 0 and 100 mm. The sum of all VAS scores and the average (H/7) values were reported [22]. SANDE was used to quantify the frequency and severity of the dry eye symptoms by placing a vertical line on a line from 1 to 100 mm to indicate how often the sensation of dry eye occurred and how severe it was. In order to demonstrate that patients had the entire range of possible perceptions of symptoms at their disposal when responding, one end of the scale represents the maximum conceivable symptom strength (i.e., 100%), and the other end represents no symptoms whatsoever (i.e., 0%) [23].

Contact lens wearers and those patients who showed abnormalities in the position and closure of the eyelids were excluded from the study.

For each enrolled patient, the following data were recorded and used for the statistical analysis: demographics; systemic and ocular history, including the modified Rodnan skin score (mRSS) and previous or current use of systemic and ocular drugs; timing, duration, frequency, and severity of the ocular symptoms; signs such as blepharitis, eyelid margin irregularity, thickening of the Marx line, lid edema, sclera/conjunctival hyperemia, corneal Oxford staining, tyndall, and any iris, lens, vitreous, or retina abnormalities; and laboratory and diagnostic findings.

### Statistical Analysis

Descriptive statistics were performed by calculating the mean and standard deviation (SD) for the continuous variables and the percentage for the discrete variables. Values from the right and left eyes were analyzed separately. The Mann–Whitney test was used to compare the continuous variables, whereas the Fisher exact test was used to compare the categorical data between the groups. The Pearson’s correlation test was used to analyze the correlations between the clinical tests and the mRSS scores. A *p*-value < 0.05 was considered statistically significant.

## 3. Results

In this retrospective study, 60 consecutive SSc patients (55 females and five males; mean age, 57 ± 14.48 years) were included. The demographic and clinical characteristics, including the mRSS and the laboratory findings, are shown in Table 1, and the ophthalmological findings are shown in Table 2.

MGD was present in 86.66% of cases (grade 3 + 4, Figure 2). An example of the severity of the involvement of the meibomian glands is shown in Figure 3.

The mean osmolarity of the tear fluid was 328.51 ± 23.8 mOsmol/L. Furthermore, a positive significant correlation was found between the osmolarity and the mRSS (r = 0.79; *p* < 0.001), showing a marked increase in the subjects with a more serious disease (Figure 4 Graph 1).

The mean Schirmer test I value was 11.04 ± 5.3 mm/5 min in SSc patients (Table 2). A negative correlation was found between the Schirmer test I (mm) and the skin score (r = −0.6; *p* < 0.01), showing a marked decrease in the subjects with a more serious disease (Figure 4 Graph 2).

The mean NIF-TBUT time resulted in 3.4 ± 3.1 s in the tears of SSc patients. A negative correlation was found between the NIF-BUT (sec) and the skin score (r = −0.76; *p* < 0.01), showing a marked decrease in the subjects with a more serious disease (Figure 4 Graph 3).

The mean LLT value was 42.95 ± 20.91 nm. A negative correlation was found between the LLT (nm) and the skin score (r = −0.85; *p* < 0.001), showing a marked decrease in the subjects with a more serious disease (Figure 4 Graph 4).

Figure 5 shows a positive correlation between the NIF-BUT and LLT (r = 0.66; *p* < 0.001).

Data regarding osmolarity, LLT, NIF-BUT, and Schirmer test I were further processed, stratifying them by age. We observed 22 women under the age of 45 (fertile age, Group 1) and 23 women over the age of 45 (postmenopausal age, Group 2). The data analysis showed a lipid tear dysfunction in both groups, without any significant difference between them (*p* = NS). The results are shown in Table 3.

The VAS mean value was 27 ± 9.3, while the SANDE frequency and severity mean values were 50.93 ± 14.26 and 48.1 ± 6.99, respectively. There was a positive correlation between the VAS and the skin score (r = 0.38; *p* < 0.05), showing marked symptoms in subjects with more a serious disease (Figure 6). On the contrary, no correlation was found between SANDE, neither the frequency nor the severity, and the skin score (frequency: r = 0.03, *p* > 0.05; severity: r = 0.16, *p* > 0.05) (Figure 6).

Interestingly, in SSc patients, a significant correlation was found between the disease duration and objective tests such as LLT, but no significant correlation with the NIF-BUT or osmolarity. For Schirmer I, no significant association was detected. The results of the correlation analysis are shown in Table 4 and Figure 7.

## 4. Discussion

Dry eye is a multifactorial condition that affects tear production and the ocular surface, resulting in symptoms of discomfort, visual disturbance, and tear film instability with potential damage to the ocular surface [24]. DES is associated with many connective tissue diseases, and it has been also reported as a clinical manifestation in the course of SSc [19]. Indeed, when the ocular immune system is excessively stimulated, and/or the immunoregulatory mechanisms are disrupted, the balance between the innate and adaptive phases is dysregulated. This results in an abnormal production of autoantibodies and cell-mediated autoimmunity response, which leads to a fibroblast dysfunction, ocular surface inflammation, and chronic DES [25].

In our retrospective study, severe DES occurred in 84% of the SSc patients, and more severely in women. The eyelids were involved in more than 86% of cases, secondary to MGD. NIF-TBUT, LLT, and Schirmer test I were also significantly reduced, as well as, accordingly, the severe skin involvement.

The use of tear osmolarity as a suitable dry eye test has already been reported in the literature. Indeed, it shows good performance in dry eye diagnosis, higher than the other tests considered, mainly in severe dry eye. Additionally, tear osmolarity values should be interpreted as an indicator of the DES evolutionary process to severity [26]. Versura et al. have already reported normal tear osmolarity values of 296.5 ± 9.8 mOsm/L. Increasing values were shown for stepwise DES severity (mild to moderate to severe dry eye, respectively: 298.1 ± 10.6 vs. 306.7 ± 9.5 vs. 314.4 ± 10.1). A progressive worsening occurred in all the parameters with an increase in the DES severity [26]. In our study, the SSc patients had a mean tear osmolarity value of 328.51 ± 23.8 mOsm/L. An increase in the osmolarity of the tear film has been reported for immunological diseases involving the eyes [25,27]; therefore, significantly higher than normal levels are reported in the literature [18,26]. Additionally, a direct correlation was found between the tear osmolarity and the skin score (mRSS).

To date, two case-control studies investigated the DES in SSc by means of Schirmer I testing, reporting the numerical values rather than applying cut-off values for differentiating normal from abnormal [10,28]. Both reported a decrease in the tear production in the SSc group, but only Atik et al. found significant results, for both the right and left eye [10]. The sub-analysis of diffuse- and limited-type SSc was not included in either report.

NIF-BUT and LLT are the two most significant parameters for the diagnosis of lipid tear dysfunction [29]. LLT represents the condition and function of the meibomian gland in meibomian gland dysfunction or dry eye [30]. LLT, blinking pattern, and meibomian gland images were assessed using the LipiView II (TearScience, Morrisville, NC, USA) ocular surface interferometer. In recent years, thanks to the development of the ocular surface interferometer, an increased number of studies regarding the tear lipid layer in dry eye patients have been published [31,32,33]. It has been hypothesized that the alteration of the lipid layer of tears caused by chronic inflammation and the dysfunction of the meibomian glands may be linked to three factors: (1) peri-glandular fibrosis, inflammation of the lacrimal glands, including the meibomian glands, linked to systemic inflammatory disease; (2) reduced mobility of the eyelids due to fibrosis of the connective tissue, which causes incomplete blinking; and (3) increased osmolarity due to greater evaporation of tears from the ocular surface, which is a consequence of the thinning of the lipid layer [28,34,35].

Considering the high prevalence of SSc among women, a potential role of hormonal factors in the pathogenesis of DES [36] may be suspected. Thus, we further processed the patients’ data, stratifying them by age. However, no significant difference in lipid tear dysfunction was found between the fertile and postmenopausal groups. These results lead us to believe that DES is closely linked to immunological disorder rather than to hormonal influence.

In our study, a high positive correlation was found for osmolarity, the subjective symptoms of DES (VAS), and the severity of SSc defined according to the skin score. Contrarily, a negative correlation was found for the NIF-BUT, LLT, and the Schirmer test I.

In general, the association between the VAS and SANDE scores as a subjective parameter and the measured objective clinical test variables was weak [19,24,37]. In our study, there was a positive correlation between the VAS and the skin score, showing marked symptoms in subjects with a more serious disease (Figure 6). On the contrary, no correlation was found between the SANDE (both frequency and severity) and the skin score. Wangkaew et al. also investigated dry eye symptoms using a dedicated questionnaire and found a significantly higher prevalence of dry eye symptoms in SSc (54%) compared to healthy controls (16%) (*p* < 0.01) [38]. However, the questionnaire included possible confounding factors, such as smoking habits and previous and current use of xerogenic drugs.

In conclusion, our study shows that routine ocular assessments in the diagnosis and follow-up of patients with SSc allows an early recognition of DES ocular manifestations and their early treatment, thus reducing patient discomfort and ocular morbidity, as well as improving the quality of life of the patient. Indeed, although the etiology differs between groups of patients with SSc and Sjogren’s syndrome, functional tests seem to be strongly influenced by individual aspects, such as age and the duration of the disease. Accordingly, tests of tear function may be crucial diagnostic procedures that could be used at an early stage, even in patients without symptoms, to grade the different spectrum of the ocular surface’s involvement and to identify an adequate systemic as well as topical therapy.

## 5. Conclusions

SSc patients show lipid tear dysfunction related to the severity and duration of the disease. In this study, significant results were reported in the correlations between the SSc severity and the main parameters for assessing the state of tear dysfunction: Schirmer test I, osmolarity, NIF-BUT, and LLT. LLT alteration is the most significative and is closely connected to disorders of the meibomian glands. Indeed, the inflammation and subsequent atrophy of the meibomian glands are constantly observed. Moreover, the VAS and SANDE questionnaires showed marked symptoms in subjects with a more serious disease. According to our results, an early ophthalmologic examination may be necessary in all cases of patients with SSc, whether or not they present ocular symptoms, to prevent the risk of serious complications.

## Figures and Tables

**Figure 1 diagnostics-10-00404-f001:**
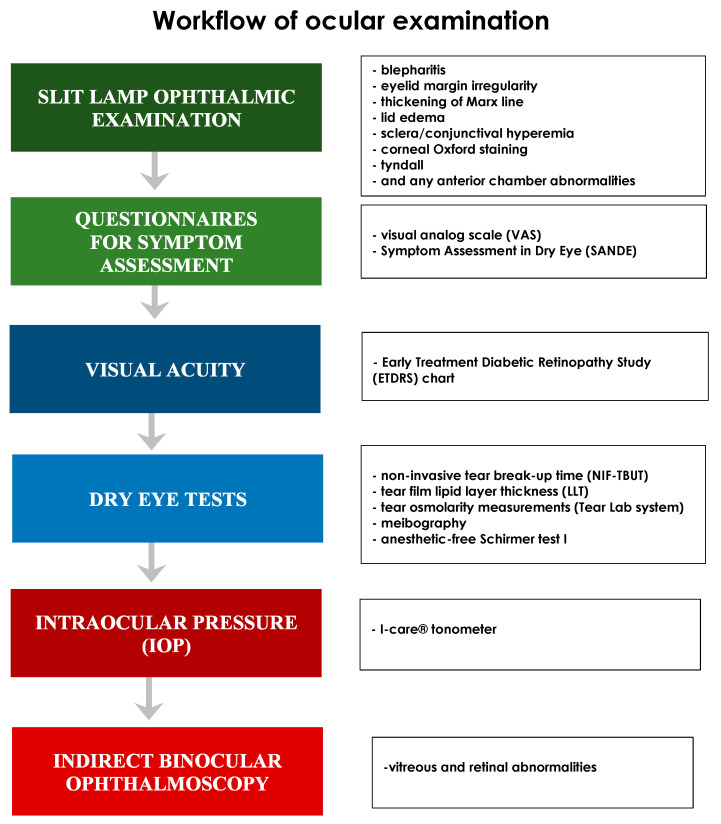
The workflow of the study. The Figure describes the sequence of ophthalmological examination performed in SSc patients enrolled in the study.

**Figure 2 diagnostics-10-00404-f002:**
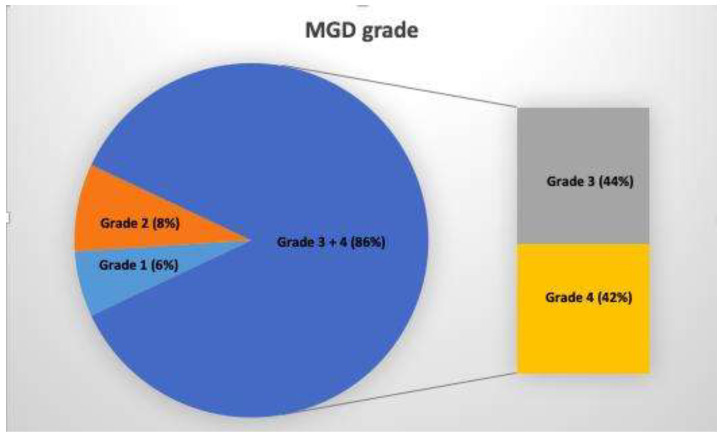
A relevant prevalence of the severe involvement of the meibomian gland was found (85% grade 3 + 4) in patients with systemic sclerosis (SSc) using meibography. MGD: meibomian gland disease.

**Figure 3 diagnostics-10-00404-f003:**
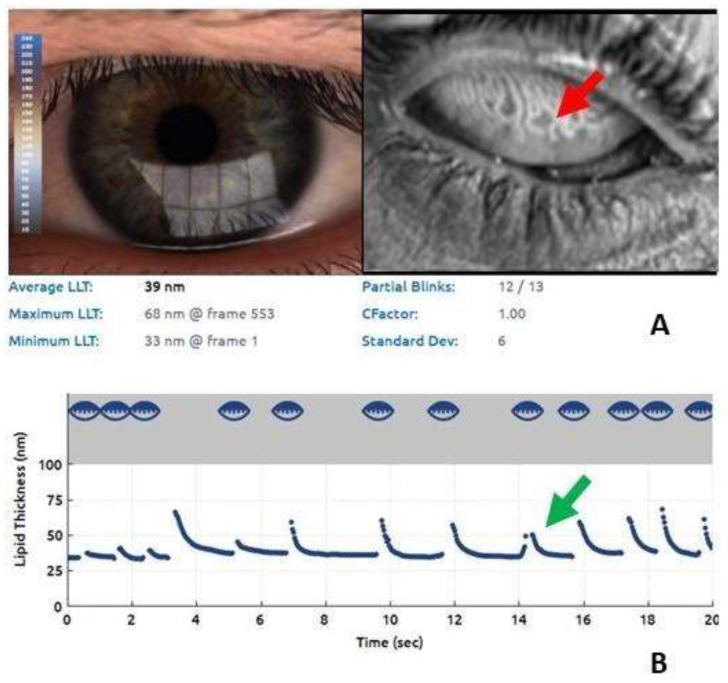
Lipid imaging report and meibography using the Lip iView II (TearScience, Morrisville, NC, USA) ocular surface interferometer. (**A**) Remarkable alterations of the meibomian glands, the dilation of their bodies, the tortuosity of the glandular ducts, and the loss of glandular tissue (blue arrow). (**B**) Significant reduction of the LLT (39 nm) (red arrow).

**Figure 4 diagnostics-10-00404-f004:**
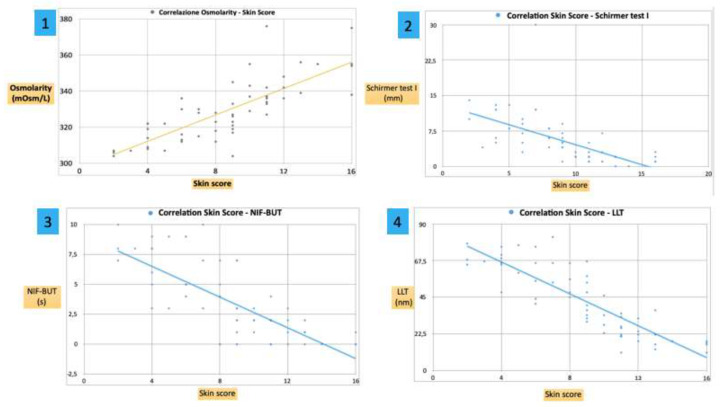
Graph 1: a positive correlation was found between osmolarity (Osmol/L) and the skin score (r = 0.79; *p* < 0.001), showing a marked increase in the subjects with a more serious disease. Graph 2, Graph 3 and Graph 4: a negative correlation was found between the Schirmer test I (mm), NIF-BUT (s), LLT (nm) and the skin score (r = −0.6, r = −0.76 and r = −0.85 with *p* < 0.01, *p* < 0.01 and *p* < 0.001 respectively), showing a marked decrease in the subjects with a more serious disease.

**Figure 5 diagnostics-10-00404-f005:**
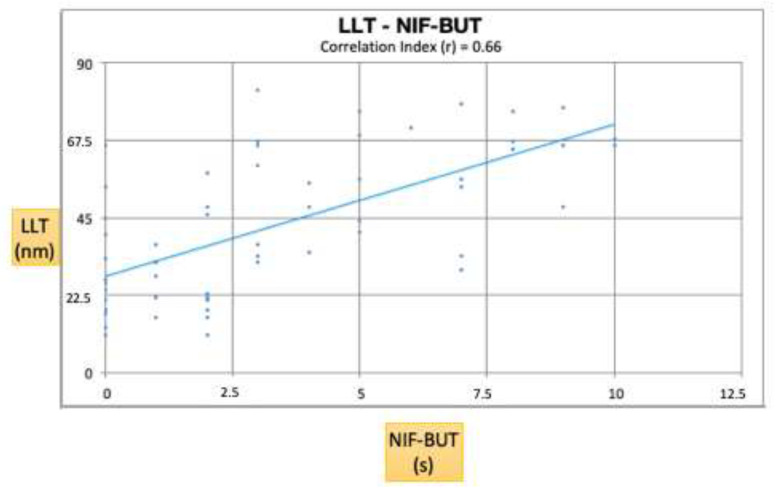
NIF-BUT and LLT are the two most significant parameters for the diagnosis of lipid tear dysfunction. This graph shows a positive correlation between these two parameters (r = 0.66; *p* < 0.001).

**Figure 6 diagnostics-10-00404-f006:**
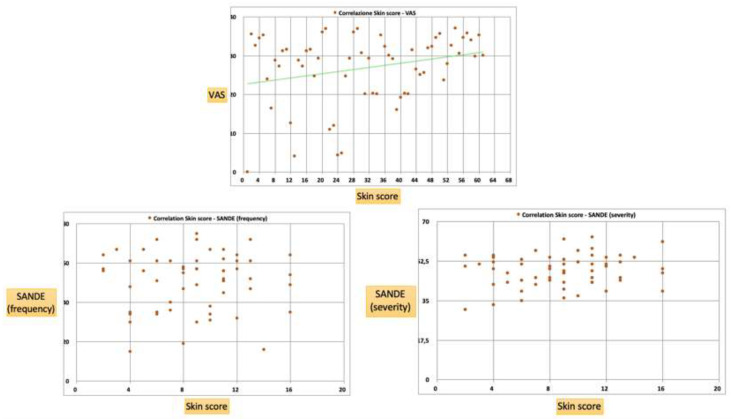
A positive correlation was found between the VAS and the skin score (r = 0.38; *p* < 0.05), showing marked symptoms in the subjects with a more serious disease. No correlation was found between the SANDE (frequency and severity) and the skin score (*p* > 0.05).

**Figure 7 diagnostics-10-00404-f007:**
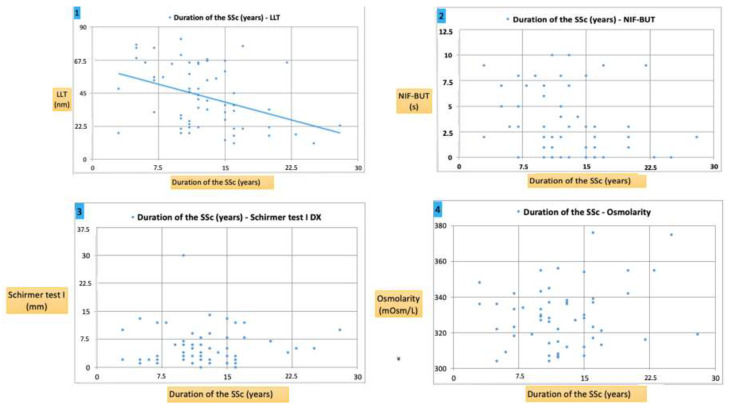
Graph 1: a negative correlation was found between the LLT (nm) and the duration of SSc (*p* < 0.05), showing a marked decrease in patients with a longer duration of SSc. Graph 2 and Graph 3: no correlation was found between the NIF-BUT (s), Schirmer test I (mm) and the duration of SSc (*p* > 0.05, both cases). Graph 4: a positive correlation was found between the osmolarity (mOsmol/L) and the duration of SSc (r = 0.2; *p* > 0.05), showing an increase in patients with a longer duration of SSc.

**Table 1 diagnostics-10-00404-t001:** Demographic and the clinical characteristics and laboratory findings.

**Demographic Characteristics**	
Age (y) (mean ± SD)	57 ± 14.48
Sex (F/M)	55/5
Race (%)	Caucasian (100)
Weight (kg) (mean ± SD)	56 ± 6.63
Height (cm) (mean ± SD)	162.71 ± 5.71
BMI (mean ± SD)	21.14 ± 2.65
**Clinical Characteristics**	
Modified Rodnan skin score (mRSS) (minimum to maximum average) (mean ± SD)	2–16 [8.86 ± 3.65]
SSc type (n) (%) • Early • Limited • Diffuse	13 [21.66%] 32 [53.33%] 26 [43.33%]
**Laboratory Findings**	
Antinuclear antibodies (ANAs)	78.33%
Anti-centromere antibodies (ACAs)	55%
Anti-Scl-70 antibodies (SCL70)	50%

BMI: body mass index; SD: standard deviation.

**Table 2 diagnostics-10-00404-t002:** Ophthalmological findings.

Ophthalmological Findings		
	*n* (%)	±SD
Meibomian glands disease	52 (86.66%)	
Blepharitis	39 (65%)	
Eyelid margin irregularity	47 (78.33)	
Marx line	48 (80%)	
Lid edema	5 (8.33%)	
Scleral/conjunctival hyperemia	15 (25%)	
Cornea (Oxford staining)	39 (65%)	
Tyndall	4 (6.66%)	
Iris	5 (8.33%)	
Lens	15 (25%)	
Vitreous	35 (58.33%)	
Retina	7 (11.66)	
Best corrected visual acuity (BCVA)	15 (25%)	
IOP (mmHg)	13.68	2.45
VAS	27	9.3
SANDE (frequency) (mean ± SD)	50.93	14.26
SANDE (severity) (mean ± SD)	48.1	6.99
Schirmer test I (mm/5 m) (mean ± SD)	11.04	5.3
NIF-TBUT (sec) (mean ± SD)	3.4	3.1
Tear osmolarity (Tear Lab) (mean ± SD)	328.51	23.8
LLT (nm) (mean ± SD)	42.95	20.91

mmHg: millimeters of mercury; m: minute; sec: second; SD: standard deviation.

**Table 3 diagnostics-10-00404-t003:** Values of lipid tear dysfunction in fertile (Group 1) and postmenopausal women (Group 2).

	Group 1 * Mean ± SD	Group 2 ** Mean ± SD	*p* = Value	
Osmolarity (mOsm/L)	338.8 ± 14.74	333.63 ± 22.65	0.94	NS
LLT (nm)	36.6 ± 21.32	44 ± 23.14	0.45	NS
NIF-BUT (s)	2.5 ± 2.71	4.09 ± 3.53	0.26	NS
Schirmer test I (mm/5 m)	4.9 ± 4.65	3.54 S ± 2.69	0.41	NS

* Women under the age of 45; ** women over the age of 45. NS: not significant.

**Table 4 diagnostics-10-00404-t004:** Linear correlation of the VAS, SANDE (frequency), SANDE (severity), Schirmer test I, osmolarity, NIF-BUT, and LLT with the skin score.

Linear Correlation Coefficient (Skin Score)				
	r	Significance	*p*	Correlation
VAS	0.38	S	<0.05	Positive
SANDE (frequency)	0.03	NS	>0.05	No correlation
SANDE (severity)	0.16	NS	>0.05	No correlation
Schirmer test I (mm)	0.6	S	<0.01	Negative
Osmolarity	0.79	S	<0.001	Positive
NIF-BUT (sec)	0.76	S	<0.01	Negative
LLT (nm)	0.85	S	<0.001	Negative

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
