# Peer review of "Dry Eye in Systemic Sclerosis Patients: Novel Methods to Monitor Disease Activity"

_diagnostics, 2020, doi:10.3390/diagnostics10060404_

Round 1
Reviewer 1 Report
In the article “Dry eye in systemic sclerosis patients: novel methods to monitor disease activity”, the authors, Gagliano C. et. al. aim to explore the characteristics in dry eye symptoms in patients with systemic sclerosis, and its possible correlation with the severity of the disease.
Below, please you will find several issues that I think need to be addressed:
1- Have the procedures performed been approved by the hospital's Ethics Committee?
2- The authors must explain in more detail how the different analyzes have been carried out in the Material and Methods section.
3- Since the authors have a greater population of women than men, have you not thought that perhaps there is a hormonal factor that interferes with the results obtained?
4- Table 1 must explain all the initials used (ANA, ACA, SCL 70).
5- In some figures such as figure 5, figure 6, figure 7, etc. The units on the axes of the correlation need to be indicated. Although indicated in the figure legend, it must be shown in the graph. All figures should be reviewed, and their presentation improved.
6- It should indicate which are the measurement ranges of the VAS and SANDE scales.
7- Regarding the osmolarity data, it indicates that there is an increase with respect to the normal values. Have they measured osmolarity in healthy subjects?
8-Has the NIF-BUT been measured in control subjects? Although you indicate that the normal ranges are between 20 and 30. Have you verified it with the same procedure with that you have measured the study patients in healthy subjects?
9- In the results section, between lines 138 and 141, indicates that there is a significant positive correlation between disease duration in SSc and NIF-BUT and osmolarity tests. Figures 11 and 13 do not show these results. For the correlation with EL NIF-BUT shows a r = 0.26 with p> 0.05 and in the correlation with osmolarity shows a r = 0.2 and p> 0.05, both results not significant.
Authors should review these results and explain them more clearly in the results section.
10- In line 209 of the discussion and later in the conclusion, the authors indicate that inflammation is observed in the meibomian glands. What result are they based on to make that statement? The authors have measured inflammatory interleukins in the tear ?. This information will help them determine the existence of an inflammatory process.
Author Response
Sunday, 24 May 2020
Dear Reviewer 1,
We appreciate the interest that you have taken in our manuscript and the constructive criticism you have given. Based on your comments, we have made changes to the manuscript, which are detailed below in italics. All modifications to the text in the manuscript are highlighted with red tracked changes. Additionally, the manuscript underwent an English editing.
Reviewer #1:
- Have the procedures performed been approved by the hospital's Ethics Committee?
Authors’ response:
Thanks for the comment. All the procedures were performed as routine clinic procedures. However, we have obtained approval from our local Ethics Committee. We have now made changes to the manuscript according to your recommendations above (from line 90 to line 93).
- The authors must explain in more detail how the different analyzes have been carried out in the Material and Methods section.
Authors’ response:
Thanks for suggestions. All the Materials and Methods section has been changed to provide more details (from line 86 to line 149). Additionally, the workflow of the study has been presented in Figure 1.
- Since the authors have a greater population of women than men, have you not thought that perhaps there is a hormonal factor that interferes with the results obtained?
Authors’ response:
We Absolutely agree, thank you for your comment.
In this retrospective study, we recruited 60 consecutive SSc patients (55 females; 5 males) (mean age 57 ± SD 14,48 years) diagnosed according to the international criteria for SSc [1,2].
SSC peak incidence is between 30 and 60 years of age with a female predilection (6:1 ratio) [3].
Hormonal factors seem to be involved in DES which is strongly correlated, in the post- menopausal period, with the reduction of estrogens, androgens and progesterone as previously reported by Gagliano et al. [4].
Thus, according to reviewer’s suggestion, we have further processed the data, stratifying them by age. We observed 22 women under the age of 45(fertile age, Group 1) and 23 women over the age of 45 (postmenopausal age, Group 2). Results are shown in Table 3.
Table 3. Values of lipid tear dysfuction in fertile (Group 1) and postmenopausal women (Group 2).
In our case series, lipid tear dysfunction was found in both fertile and postmenopausal ages without any significant difference was found between the two groups of women regarding osmolarity, LLT, NIF-BUT and Schirmer I. This result leads us to believe that the presence of DES is closely linked to the immunological disorder rather than to the influence of hormones.
These aspects concepts were reported in Introduction section (line 47), Results section (from line 247 to line 255, and Table 3), Discussion section (from line 357 to line 361).
Reference:
- LeRoy EC, Medsger TA Jr. Criteria for the classification of early systemic sclerosis. J Rheumatol. 2001; 28,1573–1576.
- Preliminary criteria for the classification of systemic sclerosis (scleroderma). Subcommittee for scleroderma criteria of the American Rheumatism Association Diagnostic and Therapeutic Criteria Committee. Arthritis Rheum. 1980;23(5),581–590.
- Kreps EO, Carton C, Cutolo M, Cutolo CA, Vanhaecke A, Leroy BP, Smith V. Ocular involvement in systemic sclerosis: A systematic literature review, it's not all scleroderma that meets the eye. Semin Arthritis Rheum. 2019 Aug;49(1):119-125. doi: 10.1016/j.semarthrit.2018.12.007.
- Gagliano C, Caruso S, Napolitano G, Malaguarnera G, Cicinelli MV, Amato R, Reibaldi M, Incarbone G, Bucolo C, Drago F, Avitabile T. Low levels of 17-β-oestradiol, oestrone and testosterone correlate with severe evaporative dysfunctional tear syndrome in postmenopausal women: a case-control study. Br J Ophthalmol. 2014 Mar;98(3):371-6. doi: 10.1136/bjophthalmol-2012-302705.
- Table 1 must explain all the initials used (ANA, ACA, SCL 70).
Authors’ response:
Thanks for the comment. We have now added the explanation of the initials used.
- In some figures such as figure 5, figure 6, figure 7, etc. The units on the axes of the correlation need to be indicated. Although indicated in the figure legend, it must be shown in the graph. All figures should be reviewed, and their presentation improved.
Authors’ response:
Thank you for your comments. We have reviewed all Figures according to your suggestions.
Additionally, Figure 2 that shows a Lipid Imaging Report and Meibography with Lip iView II (TearScience, Morrisville, NC, USA) ocular surface interferometer has been added.
- It should indicate which are the measurement ranges of the VAS and SANDE scales.
Authors’ response:
Thank you for the suggestion. We have now specified the measurement ranges of the VAS and SANDE scales (from line 129 to line 140).
Visual analogue scales (VAS) was the measuring instrument for the documentation of symptoms (Foreign body sensation, Burning/Tingling, Itching, Pain, Sticking sensation, Blurred vision, Photophobia). Intensity of the discomfort is indicated with a vertical line on the line between 0- and 100-mm. Sum of all VAS and average (H/7) were reported reported [1].
Symptom Assessment iN Dry Eye (SANDE) was the method used to quantify the frequency and severity of dry eye symptoms by placing a vertical line on a line from 1 to 100 mm to indicate how often the sensation of dry eye occurs and how severe it is. In order to demonstrate that patients have the entire range of possible perceptions of symptoms at their disposal when responding, one end of the scale represents the maximum conceivable symptom strength (i.e., 100%), the other end no symptoms whatsoever (i.e., 0%) [2].
References:
- Ludger Klimek, et al. Visual analogue scales (VAS): Measuring instruments for the documentation of symptoms and therapy monitoring in cases of allergic rhinitis in everyday health care. Allergo J Int. 2017; 26(1): 16– 24. doi:10.1007/s40629-016-0006-7.
- Francisco Amparo, et al. Comparison of Two Questionnaires for Dry Eye Symptom Assessment: The Ocular Surface Disease Index and the Symptom Assessment in Dry Eye. Ophthalmology 2015 Jul;122(7):1498-503. doi: 10.1016/j.ophtha.2015.02.037.
- Regarding the osmolarity data, it indicates that there is an increase with respect to the normal values. Have they measured osmolarity in healthy subjects?
Authors’ response:
Thanks for the comment.
The use of tear osmolarity as a suitable dry eye test has been already reported in the literature. Indeed, it showed a good performance in dry eye diagnosis, higher than the other tests considered, mainly in severe dry eye. Additionally, tear osmolarity values should be interpreted as an indicator of DED evolutionary process to severity.
Versura et al.* reported tear osmolarity normal values of 296.5 +/- 9.8 mOsm/L. Increasing values were shown stepwise DE severity (mild to moderate to severe dry eye, respectively: 298.1 +/- 10.6 vs. 306.7 +/- 9.5 vs. 314.4 +/- 10.1, p < 0.05). A progressive worsening occurred in all the parameters with DED severity increase.
In our experience we have already tested both in healthy group and pathological group for different diseases the accuracy and accuracy and repeatability of osmolarity measurements. In 48 healthy patients we found an osmolarity mean value of 301,45 ± 13,34 mOsm/L (range between 289 and 308 mOsm/L). Considering these results, the difference with the SSc patients (328,51 ± 23,8 mOsm/L) was statistically significant.
These aspects have been added in the Discussion section.
Reference:
*- P Versura, V Profazio, E C Campos. Performance of Tear Osmolarity Compared to Previous Diagnostic Tests for Dry Eye Diseases Curr Eye Res. 2010 Jul;35(7):553-64. doi: 10.3109/02713683.2010.484557.
- Has the NIF-BUT been measured in control subjects? Although you indicate that the normal ranges are between 20 and 30. Have you verified it with the same procedure with that you have measured the study patients in healthy subjects?
Authors’ response:
Thanks for the comment.
The equipment that we used for the NIF-BUT has a regulatory database to which reference is made by means of a color scale.
Unluckily, our study has the limitation to be a retrospective study without a control group. Anyway, we have verified the procedure in a prospective study of 30 healthy subjects and the values obtained range from 15 to 26 (mean 20,43±3,4 SD).
- In the results section, between lines 138 and 141, indicates that there is a significant positive correlation between disease duration in SSc and NIF-BUT and osmolarity tests. Figures 11 and 13 do not show these results. For the correlation with EL NIF-BUT shows a r = 0.26 with p> 0.05 and in the correlation with osmolarity shows a r = 0.2 and p> 0.05, both results not significant.
Authors’ response:
Thanks for the comment.
In Figure 11 the correlation is negative and not positive how it is written in the text. In the text it has been corrected. Furthermore, as you have noted, the two correlations data (Pearson correlation coefficient) between the duration of the SSc and NIF-BUT and Osmolarity are not significant. Authors reviewed these results and explain them more clearly in the results section. Figure 11 has been changed now in Figure 12.
- In line 209 of the discussion and later in the conclusion, the authors indicate that inflammation is observed in the meibomian glands. What result are they based on to make that statement? The authors have measured inflammatory interleukins in the tear? This information will help them determine the existence of an inflammatory process.
Authors’ response:
Thanks for the comment.
The inflammation of the meibomian glands was assessed by observing the alteration of the eyelid margin, the thickening of the Marx line, the morphological alteration of the glandular bodies at meibography. This aspect has been reported in the Methods section (from line 121 to line 123).
In the literature [*] also the dosage of MMP9 in tears by Inflamma-dry test has been reported to test values of inflammatory interleukins in the tear. However, given the retrospective nature of the study and its lack in the routine ocular assessment, data on the dosage of MMP9 are not available.
The conclusions and the discussion section have been revised.
Reference:
* Nicole Lai Lanza, Felipe Valenzuela, Victor L Perez, and Anat Galor. The Matrix Metalloproteinase 9 Point-of-Care Test in Dry Eye. Ocul Surf. 2016 Apr; 14(2): 189–195. doi: 10.1016/j.jtos.2015.10.004
Looking forward to hearing from you.
Sincerely yours,
All coauthors

Reviewer 2 Report
The authors have presented data to show the novel method to monitor the dry eye in systemic sclerosis patients. Although the paper is interesting shows a simple method to study how clinical parameters could be used to study the DES disease activity in SSc patients.
But the background section of abstract says study was designed to for establishing correlation between SSc and DES.. is this study about exploring correlation of various diagnostic associated with DES in SSc patient to study the severity of ophthalmological disorders? or it is about correlating SSc with DES? Please make changes in your abstract accordingly...
This paper can be accepted after the following taking following suggestions in account.
- The introduction lacks a connection between various parts. The link between Systemic sclerosis and Dry eye syndrome (DES).. please address why did you think to focus on dry eye syndrome in systemic sclerosis condition is important? I think author tried to write about this in introduction line 64-67. But it is unclear and incomplete.
- The introduction is too complex and does not provide a clear reason for what is the purpose of this diagnostic work. It does not answer why authors performed statistical analysis to understand the severity of the disease. And what is the significance of the analytical approach (used in this paper) in the diagnostic world for DES or any other disease condition to monitor disease progression?
- Any specific reason for selecting patients with the average age group of 55? Does this systemic sclerosis leading to DES happen only happens in elderly patients, please discuss this in the introduction section?
- Please provide a schematic to show the workflow for the study for ease of understanding.
- Discussion is poorly framed.. for example... a) it has parts of conclusion in it.. for example line 207 to 211. b) discussion does not provide an explanation of the results and their addition to the existing pool of literature.
- It will be good to have correlation based statistical data in table for ease of understanding of the results and its explanation in text.
- Conclusion is incomplete. Please suggests what parameters showed significant or insignificant effect in the ophthalmologic evaluations (relating to DES) in SSc patients.
Author Response
Sunday, 24 May 2020
Dear Reviewer 2,
We appreciate the interest that you have taken in our manuscript and the constructive criticism you have given. Based on your comments, we have made changes to the manuscript, which are detailed below in italics. All modifications to the text in the manuscript are highlighted with red tracked changes. Additionally, the manuscript underwent an English editing.
Reviewer #2:
1- The authors have presented data to show the novel method to monitor the dry eye in systemic sclerosis patients. Although the paper is interesting shows a simple method to study how clinical parameters could be used to study the DES disease activity in SSc patients.
But the background section of abstract says study was designed to for establishing correlation between SSc and DES... is this study about exploring correlation of various diagnostic associated with DES in SSc patient to study the severity of ophthalmological disorders? or it is about correlating SSc with DES? Please make changes in your abstract accordingly...
Authors’ response:
Thank you for your positive feedback and valuable comments. As reported in the Introduction section, the aim of this study is to explore the prevalence and characteristics in ocular symptoms and signs in patients with SSc and to establish any correlation with the severity of the disease using different diagnostic tools. According to your suggestion, we have changed the background section of the abstract and improved its flow (from line 25 to line 27).
This paper can be accepted after the following taking following suggestions in account.
- The introduction lacks a connection between various parts. The link between Systemic sclerosis and Dry eye syndrome (DES)… please address why did you think to focus on dry eye syndrome in systemic sclerosis condition is important? I think author tried to write about this in introduction line 64-67. But it is unclear and incomplete.
Authors’ response:
Thank you for your positive feedback and valuable comments that gave us possibility to improve the flow of the Indroduction section. We have better clarify the reason why to focus on DES-related findings in SSc patients is crucial, considering the lack of a strong evidence reported in literature (from line 68 to line 81). Indeed, while a multisystemic involvement is well and widely documented in SSc patients, to date, only single case reports or small case studies regarding the ocular involvement in SSC patients have been reported in the literature. Thus, further evidence is demanded.
- The introduction is too complex and does not provide a clear reason for what is the purpose of this diagnostic work. It does not answer why authors performed statistical analysis to understand the severity of the disease. And what is the significance of the analytical approach (used in this paper) in the diagnostic world for DES or any other disease condition to monitor disease progression?
Authors’ response:
As already reported in the replay to point 1, we have rewritten the introduction section according to reviewer’s suggestions, thus simplifing its flow. Additionally, the purpose of the study has been better clarify (from line 68 to line 81).
We like underline how an analytical approach to monitor the progression of the dry eye condition and the SSc has allowed us to find quickly an effective solution to avoid more serious consequences for the eye and for the body in general of SSc patients.
Furthermore, to analyze the situation from multiple points of view offers to Ophthalmologists the possibility of making decisions on the most appropriate therapies, both systemic and topical.
- Any specific reason for selecting patients with the average age group of 55? Does this systemic sclerosis leading to DES happen only happens in elderly patients, please discuss this in the Introduction section?
Authors’ response:
We Absolutely agree, thank you for your comment.
In this retrospective study, we recruited 60 consecutive SSc patients (55 females; 5 males) (mean age 57 ± SD 14,48 years) diagnosed according to the international criteria for SSc [1,2].
SSC peak incidence is between 30 and 60 years of age with a female predilection (6:1 ratio) [3].
Hormonal factors seem to be involved in DES which is strongly correlated, in the post- menopausal period, with the reduction of estrogens, androgens and progesterone as previously reported by Gagliano et al. [4].
Thus, according to reviewer’s suggestion, we have further processed the data, stratifying them by age. We observed 22 women under the age of 45(fertile age, Group 1) and 23 women over the age of 45 (postmenopausal age, Group 2). Results are shown in Table 3.
Table 3. Values of lipid tear dysfuction in fertile (Group 1) and postmenopausal women (Group 2).
*women under the age of 45
**women over the age of 45
In our case series, lipid tear dysfunction was found in both fertile and postmenopausal ages without any significant difference was found between the two groups of women regarding osmolarity, LLT, NIF-BUT and Schirmer I. This result leads us to believe that the presence of DES is closely linked to the immunological disorder rather than to the influence of hormones.
These aspects concepts were reported in Introduction section (line 47), Results section (from line 247 to line 255, and Table 3), Discussion section (from line 357 to line 361).
Reference:
- LeRoy EC, Medsger TA Jr. Criteria for the classification of early systemic sclerosis. J Rheumatol. 2001; 28,1573–1576.
- Preliminary criteria for the classification of systemic sclerosis (scleroderma). Subcommittee for scleroderma criteria of the American Rheumatism Association Diagnostic and Therapeutic Criteria Committee. Arthritis Rheum. 1980;23(5),581–590.
- Kreps EO, Carton C, Cutolo M, Cutolo CA, Vanhaecke A, Leroy BP, Smith V. Ocular involvement in systemic sclerosis: A systematic literature review, it's not all scleroderma that meets the eye. Semin Arthritis Rheum. 2019 Aug;49(1):119-125. doi: 10.1016/j.semarthrit.2018.12.007.
- Gagliano C, Caruso S, Napolitano G, Malaguarnera G, Cicinelli MV, Amato R, Reibaldi M, Incarbone G, Bucolo C, Drago F, Avitabile T. Low levels of 17-β-oestradiol, oestrone and testosterone correlate with severe evaporative dysfunctional tear syndrome in postmenopausal women: a case-control study. Br J Ophthalmol. 2014 Mar;98(3):371-6. doi: 10.1136/bjophthalmol-2012-302705.
- Please provide a schematic to show the workflow for the study for ease of understanding.
Authors’ response:
As suggested, we have added a schematic to show the workflow for the study (Figure 1). We hope it will provide a easier understanding of the study.
Additionally, Figure 2 that shows a Lipid Imaging Report and Meibography with Lip iView II (TearScience, Morrisville, NC, USA) ocular surface interferometer has been added.
Figure 1. The workflow of the study.
- Discussion is poorly framed…for example...
- a) it has parts of conclusion in it… for example line 207 to 211.
- b) discussion does not provide an explanation of the results and their addition to the existing pool of literature.
Authors’ response:
Thanks for your suggestions.
- a) The paragraph (from line 207 to 211) has been removed because it is already included in the Conclusions section.
- b) The discussioni has been improved adding the existing pool of literature available to
- It will be good to have correlation based statistical data in table for ease of understanding of the results and its explanation in text.
Authors’ response:
According to your suggestions, the Results section has benn rewritten to simplify the main findings. Additionally a table for main data correlations has been added in Table 4 and 5.
Table 4. Linear correlation of VAS, SANDE (fequency), SANDE (severity), Schirmer test I, Osmolarity, NIF_BUT and LLT with Skin Score.
Linear correlation coefficient (Skine score) |
||||
|
r |
Significance |
p |
Correlation |
VAS |
0,38 |
S |
< 0.05 |
Positive |
SANDE (frequency) |
0,03 |
NS |
>0.05 |
No correlation |
SANDE (severity) |
0,16 |
NS |
>0.05 |
No correlation |
Schirmer test I (mm) |
0,6 |
S |
<0.01 |
Negative |
Osmolarity |
0,79 |
S |
<0.001 |
Positive |
NIF-BUT (sec) |
0,76 |
S |
<0.01 |
Negative |
LLT (nm) |
0,85 |
S |
<0.001 |
Negative |
Table 5. Linear correlation of Schirmer test I, Osmolarity, NIF-BUT and LLT with duration of SSc.
Linear correlation coefficient (Duration of SSc) |
||||
|
r |
Significance |
p |
Correlation |
Schirmer test I (mm) |
0,02 |
NS |
>0.05 |
No correlation |
Osmolarity |
0,2 |
NS |
>0.05 |
Positive |
NIF-BUT (sec) |
0,26 |
NS |
>0.05 |
No correlation |
LLT (nm) |
0,4 |
S |
<0.05 |
Negative |
- Conclusion is incomplete. Please suggests what parameters showed significant or insignificant effect in the ophthalmologic evaluations (relating to DES) in SSc patients.
Authors’ response:
Thank you for your valuable comment. We have changed the conclusion according to your suggestions.
Looking forward to hearing from you.
Sincerely yours,
All coauthors

Round 2
Reviewer 1 Report
I agree with the changes made, although the graphics design could be improved.
Author Response
Tuesday, 2 June 202
Dear Reviewer 1,
We appreciate the interest that you have taken in our manuscript and the constructive criticism you have given. Based on your comments, we have made changes to the manuscript, which are detailed below in italics. All modifications to the text in the manuscript are highlighted with red tracked changes.
Reviewer’s comment:
1) I don't feel qualified to judge about the English language and style.
Authors’ replay:
The manuscript has undergone English language editing by MDPI. The text has been checked for correct use of grammar and common technical terms, and edited to a level suitable for reporting research in a scholarly journal by experienced, native English-speaking editors and according MDPI journals guidelines. Full details of the editing service can be found at https://www.mdpi.com/authors/english.
2) I agree with the changes made, although the graphics design could be improved.
Authors’ replay:
As suggested, the graphics design of Figure 1 has been improved and all tables have been professionally reviewed according to the notes done by the MDPI editorial service.
Looking forward to hearing from you.
Sincerely yours,
All coauthors
Reviewer 2 Report
I think the authors have duly answered all the comments.
But there is the minor thing figure 1 is more of pointers. It will be good if authors could make it in the form of a schematic flow diagram instead of simply providing pointers.
Author Response
Tuesday, 2 June 2020
Dear Reviewer 2,
We appreciate the interest that you have taken in our manuscript and the constructive criticism you have given. Based on your comments, we have made changes to the manuscript, which are detailed below in italics. All modifications to the text in the manuscript are highlighted with red tracked changes.
Reviewer’s comment:
1) Moderate English changes required
Authors’ replay:
The manuscript has undergone English language editing by MDPI. The text has been checked for correct use of grammar and common technical terms, and edited to a level suitable for reporting research in a scholarly journal by experienced, native English-speaking editors and according MDPI journals guidelines. Full details of the editing service can be found at https://www.mdpi.com/authors/english.
2) I think the authors have duly answered all the comments. But there is the minor thing figure 1 is more of pointers. It will be good if authors could make it in the form of a schematic flow diagram instead of simply providing pointers.
Authors’ replay:
As suggested, the graphics design of Figure 1 has been changed in the form of a flow-diagram. Additionally, all tables have been professionally reviewed and improved according to the notes done by the MDPI editorial service.
Looking forward to hearing from you.
Sincerely yours,
All coauthors